# No Training Required: Exploring Random Encoders for Sentence Classification

**John Wieting**[*]
Carnegie Mellon University
jwieting@cs.cmu.edu

**Douwe Kiela**
Facebook AI Research
dkiela@fb.com

## Abstract

We explore various methods for computing sentence representations from pre-trained word embeddings *without any training*, i.e., using nothing but random parameterizations. Our aim is to put sentence embeddings on more solid footing by 1) looking at how much modern sentence embeddings gain over random methods—as it turns out, surprisingly little; and by 2) providing the field with more appropriate baselines going forward—which are, as it turns out, quite strong. We also make important observations about proper experimental protocol for sentence classification evaluation, together with recommendations for future research.

## 1 Introduction

A sentence embedding is a vector representation of the meaning of a sentence, often created by a transformation of word embeddings through a composition function. This function is often non-linear and recurrent in nature, and usually the word embeddings are initialized with pre-trained embeddings. Well-known examples of sentence embeddings include SkipThought (Kiros et al., 2015) and InferSent (Conneau et al., 2017). The purpose of sentence embeddings is to have the meaning of a sentence encoded into a compact representation where it is readily available for downstream applications. For instance it could provide features for training a classifier or a distant metric could be applied to two representations to compute the semantic similarity of their corresponding sentences. Sentence embeddings can be trained with either an unsupervised or supervised objective, and they are often evaluated using transfer tasks, where a ahallow classifier is trained on top of the learned sentence encoder (which is kept fixed). There has been a lot of recent interest in trying to understand better what these sentence embeddings learn (Adi et al., 2016; Linzen et al., 2016; Conneau et al., 2018; Zhu et al., 2018).

The natural language processing community does not yet have a clear grasp on the relationship between word and sentence embeddings: it is unclear how much trained sentence-encoding architectures improve over the raw word embeddings, and what aspect of such architectures is responsible for any improvement. Indeed, state-of-the-art word embeddings on their own perform quite well with simple pooling mechanisms, as reported by Wieting et al. (2015); Arora et al. (2017) and Shen et al. (2018). Given the tremendous pace of research on sentence representations, it is important to establish solid baselines for others to build on.

It has been observed that bidirectional LSTMs with max-pooling perform surprisingly well even without any training whatsoever (Conneau et al., 2017; 2018), leading to claims that such architectures "encode priors that are intrinsically good for sentence representations" (Conneau et al., 2018), similar to convolutional networks for images (Ulyanov et al., 2017). Inspired by these observations, we propose to examine the following question: given a set of word embeddings, how can we maximize classification accuracy on the transfer tasks *without any training*, i.e. without updating any parameters except for those in the transfer task-specific linear classifier trained on top of the representation. SkipThought famously took around one month to train, while InferSent requires large amounts of annotated data—we examine to what extent we can match the performance of these systems by exploring different ways for combining nothing but the pre-trained word embeddings.[1]

---

[*]Work done as an intern at Facebook AI Research.

[1]Code available at https://github.com/facebookresearch/randsent.

We go down a well-paved avenue of exploration in the machine learning research community, and exploit an insight originally due to Cover (1965): "A complex pattern-classification problem, cast in a high-dimensional space nonlinearly, is more likely to be linearly separable than in a low-dimensional space, provided that the space is not densely populated." That is, we examine three types of models for obtaining randomly computed sentence representations from pre-trained word embeddings: bag of random embedding projections, randomly initialized recurrent networks and echo state networks.

Our goal is not to obtain a new state of the art, but to put current state of the art methods on more solid footing by 1) looking at how much they gain compared to random methods; and 2) providing the field with more solid baselines going forward. We make several important observations about proper experimental protocol for sentence classification evaluation; and finish with a list of take-away recommendations.

## 2    RELATED WORK

Sentence embeddings are receiving a lot of attention. Many approaches have been proposed, varying in their use of both training data and training objectives. Methods include autoencoders (Socher et al., 2011; Hill et al., 2016) and other learning frameworks using raw text (Le & Mikolov, 2014; Pham et al., 2015; Jernite et al., 2017; Pagliardini et al., 2017), a collection of books (Kiros et al., 2015), labelled entailment corpora (Conneau et al., 2017), image-caption data (Kiela et al., 2017), raw text labelled with discourse relations (Nie et al., 2017), or parallel corpora (Wieting & Gimpel, 2017). Multi-task combinations of these approaches (Subramanian et al., 2018; Cer et al., 2018) have also been proposed. Progress has been swift, but lately we have started to observe some troubling trends in how research is conducted, in particular with respect to properly identifying the sources of empirical gains (see also Lipton & Steinhardt (2018)).

There was an issue with non-standard evaluation methods, for which SentEval (Conneau & Kiela, 2018) and then GLUE (Wang et al., 2018) were created. One often overlooked aspect of sentence representation evaluation, for example, is that logistic regression classifiers and multi-layer percep-trons (MLP) are not the same thing. To single out an example, the recent paper by Shen et al. (2018), which aims to "give baselines more love", does not compare against LSTMs with the exact same pre-processing and range of hyperparameters, in effect ignoring its own baselines, and uses a custom designed MLP, sweeping over many hyperparameters unique to their setup with different embedding dimensionsionalities.

Even when comparing InferSent and SkipThought, it is not entirely clear where differences come from: the better pre-trained word embeddings; the different architecture; the different objective; the layer normalization—e.g. what would happen if we trained a bidirectional LSTM with max-pooling using GloVe embeddings (i.e., InferSent's architecture) with a SkipThought objective or added layer normalization to InferSent? The nowadays surprisingly poor performance of the models in Hill et al. (2016) can at least partly be explained because 1) they use poorer (older) word embeddings; and 2) FastSent sentence representations are of the same dimensionality as the input word embeddings, while they are compared in the same table to much higher-dimensional representations. Obviously, a logistic regression classifier on top of a higher-dimensional input has more parameters too, giving such models an unfair advantage. In part, doing such in-full comparisons is simply not feasible, and often not appreciated by reviewers anyway, so we can hardly blame the authors of these papers. That said, we wholeheartedly agree that baselines need more love, befitting a good tradition in NLP (Wang & Manning, 2012): with this work we hope to establish even stronger baselines for future work and try to estimate how much performance is being added by training sentence embeddings on top of pre-trained word embeddings.

There has been a lot of recent interest in trying to understand what linguistic knowledge is encoded in word and sentence embeddings, for instance in machine translation (Belinkov et al., 2017; Sen-nrich, 2016; Dalvi et al., 2017), with a focus on evaluating RNNs or LSTMs (Linzen et al., 2016; Hupkes et al., 2018) or even sequence-to-sequence models (Lake & Baroni, 2018). Various probing tasks (Ettinger et al., 2016; Adi et al., 2016; Conneau et al., 2018) were designed to try to understand what you can "cram into a vector" for representing sentence meaning. We show that a lot of information may be crammed into vectors using randomly parameterized combinations of pre-trained word

embeddings: that is, most of the power in modern NLP systems is derived from having high-quality word embeddings, rather than from having better encoders.

The idea of using random weights is almost as old as neural networks, ultimately going back to ideas in multi-layer perceptrons with fixed randomly initialized first layers (Gamba et al., 1961; Borsellino & Gamba, 1961; Baum, 1988), or what Minsky and Papert call Gamba perceptrons (Minsky & Papert, 2017). The idea of fixing a subset of the network was made more explicit in (Schmidt et al., 1992; Pao et al., 1994), which some people have started to call extreme learning machines (Huang et al., 2006).[2]

Random features in machine learning are often used for low-rank approximation (Vempala, 2005), as per the Johnson-Lindenstrauss lemma; exploiting the useful properties of random matrices (Mehta, 2004). Random "kitchen sink" features have become a seminal approach in the machine learning literature (Rahimi & Recht, 2008; 2009). Similar ideas underlie e.g. double-stochastic gradient methods (Dai et al., 2014). In fact, it is well-known that random weights do well, as for example shown in computer vision with respect to convnets (Jarrett et al., 2009; Saxe et al., 2011). Similarly, the importance of random initializations has been examined in depth (Sutskever et al., 2013). In our case, we use random projections for higher-rank feature expansion of low-rank dense pre-trained word embeddings, exploiting Cover's theorem (Cover, 1965). An encoder like this does not require any training, unlike other sentence encoders such as SkipThought and InferSent. Comparing those methods to our random sentence encoders provides valuable insight into how much of a performance improvement we have actually gained from training for a long time (in the case of SkipThought) or training on expensive annotated data (in the case of InferSent which is trained on the Stanford Natural Language Inference (SNLI) Corpus Bowman et al. (2015), a large textual entailment dataset.).

The same idea of using fixed random computations underlies reservoir computing (Lukoševičius & Jaeger, 2009) and echo-state networks (Jaeger, 2001). In reservoir computing, inputs are fed into a fixed, random, *dynamical system* called a *reservoir* that maps the input into a high dimensional space. Then a trainable linear transformation of this high dimensional space is learned to predict some output signal. Echo-state networks are a specific type of reservoir computing and are further described in Section 3.1.3.

Reservoir computing has been used previously in Natural Language Processing (NLP), though it is not common in the literature. (Frank, 2006a;b) investigated whether ESNs are capable of displaying the systematicity in natural language. (Tong et al., 2007; Hinaut & Dominey, 2012) investigated the ability of ESNs for learning grammatical structure. Lastly, Daubigney et al. (2013) used ESNs to find efficient teaching strategies.

## 3 APPROACH

In this paper, we explore three architectures that produce sentence embeddings from pre-trained word embeddings, without requiring any training of the encoder itself. These sentence embeddings are then used as features for a collection of downstream tasks. The downstream tasks are all trained with a logistic regression classifier using the default settings of the SentEval framework (Conneau & Kiela, 2018). The parameters of this classifier are the only ones that are updated during training (see Section 3.2 below).

### 3.1 RANDOM SENTENCE ENCODERS

We are concerned with obtaining a good sentence representation $\mathbf{h}$ that is computed using some function $f$ parameterized by $\theta$ over pre-trained input word embeddings $\mathbf{e} \in L$, i.e. $\mathbf{h} = f_\theta(\mathbf{e}_1, \dots, \mathbf{e}_n)$ where $\mathbf{e}_i$ is the embedding for the $i$-th word in a sentence of length $n$. Typically, sentence encoders learn $\theta$, after which it is kept fixed for the transfer tasks. InferSent represents a sentence as $f = \max(\mathrm{BiLSTM}(\mathbf{e}_1, \dots, \mathbf{e}_n))$ and optimizes the parameters using a supervised cross-entropy objective for predicting one of three labels from a combination of two sentence representations: entailment, neutral or contradictive. SkipThought represents a sentence as $f = \mathrm{GRU}_n(\mathbf{e}_1, \dots, \mathbf{e}_n)$, with the objective of being able to *decode* the previous and next utterance using negative log-likelihood from the final (i.e., $n$-th) hidden state.

---

[2]See `http://elmorigin.wixsite.com/originofelm` for an interesting discussion of ELM.

InferSent requires large amounts of expensive annotation, while SkipThought takes a very long time to train. Here, we examine different ways of parameterizing $f$ for representing the sentence meaning, *without any training of* $\theta$. This means we do not require any labels for supervised training, nor do we need to train the sentence encoder for a long time with an unsupervised objective. We experiment with three methods for computing $\mathbf{h}$: Bag of random embedding projections, Random LSTMs, and Echo State Networks. In this section, we describe the methods in more detail. In the following sections, we show that our methods lead to surprisingly good results, shedding new light on sentence representations, and establishing strong baselines for future work.

### 3.1.1 BAG OF RANDOM EMBEDDING PROJECTIONS (BOREP)

The first family of architectures we explore consists of simply applying a single random projection in a standard bag-of-words (or more accurately, bag-of-embeddings) model. We randomly initialize a matrix $W \in \mathbb{R}^{D \times d}$, where $D$ is the dimension of the projection and $d$ is the dimension of our input word embedding. The values for the matrix are sampled uniformly from $[-\frac{1}{\sqrt{d}}, \frac{1}{\sqrt{d}}]$, which is a standard initialization heuristic used in neural networks (Glorot & Bengio, 2010). The sentence representation is then obtained as follows:

$$\mathbf{h} = f_{pool}(W\mathbf{e}_i),$$

where $f_{pool}$ is some pooling function, e.g. $f_{pool}(x) = \sum(x)$, $f_{pool}(x) = \max(x)$ (max pooling) or $f_{pool}(x) = |x|^{-1} \sum(x)$ (mean pooling). Optionally, we impose a nonlinearity $max(0, \mathbf{h})$. We experimented with imposing positional encoding for the word embeddings, but did not find this to help.

### 3.1.2 RANDOM LSTMS

Following InferSent, we employ bidirectional LSTMs, but in our case without any training. Conneau et al. (2017) reported good performance for the random LSTM model on the transfer tasks. The LSTM weight matrices and their corresponding biases are initialized uniformly at random from $[-\frac{1}{\sqrt{d}}, \frac{1}{\sqrt{d}}]$, where $d$ is the hidden size of the LSTM. In other words, the architecture here is the same as that of InferSent modulo the type of pooling used:

$$\mathbf{h} = f_{pool}(\text{BiLSTM}(\mathbf{e}_1, \ldots, \mathbf{e}_n)).$$

### 3.1.3 ECHO STATE NETWORKS

Echo State Networks (ESNs) (Jaeger, 2001) were primarily designed for sequence prediction problems, where given a sequence $X$, we predict a label $y$ for each step in the sequence. The goal is to minimize the error between the predicted $\hat{y}$ and the target $y$ at each timestep. Formally, an ESN is described using the following update equations:

$$\tilde{\mathbf{h}}_i = f_{pool}(W^i \mathbf{e}_i + W^h \mathbf{h}_{i-1} + b^i)$$

$$\mathbf{h}_i = (1 - \alpha)\mathbf{h}_{i-1} + \alpha\tilde{\mathbf{h}}_i,$$

where $W^i$, $W^h$, and $b^i$ are randomly initialized and are not updated during training. The parameter $\alpha \in (0, 1]$ governs the extent to which the previous state representation is allowed to *leak* into the current state. The only learned parameters in an ESN are the final weight matrix, $W^o$ and corresponding bias $b^o$, which are together used to compute a prediction $\hat{y}_i$ for the $i$th label $y_i$:

$$\hat{y}_i = W^o[\mathbf{e}_i; \mathbf{h}_i] + b^o.$$

We diverge from the typical per-timestep ESN setting, and instead use the ESN to produce a random representation of a sentence. We use a bidirectional ESN, where the reservoir states, $\mathbf{h}_i$, are concatenated for both directions. We then pool over these states to obtain the sentence representation:

| Model | Dim | MR | CR | MPQA | SUBJ | SST2 | TREC | SICK-R | SICK-E | MRPC | STSB |
|---|---|---|---|---|---|---|---|---|---|---|---|
| BOE | 300 | 77.3(.2) | 78.6(.3) | 87.6(.1) | 91.3(.1) | 80.0(.5) | 81.5(.8) | 80.2(.1) | 78.7(.1) | 72.9(.3) | 70.5(.1) |
| BOREP | 4096 | 77.4(.4) | 79.5(.2) | 88.3(.2) | 91.9(.2) | 81.8(.4) | **88.8(.3)** | 85.5(.1) | 82.7(.7) | 73.9(.4) | 68.5(.6) |
| RandLSTM | 4096 | 77.2(.3) | 78.7(.5) | 87.9(.1) | 91.9(.2) | 81.5(.3) | 86.5(1.1) | 85.5(.1) | 81.8(.5) | **74.1(.5)** | 72.4(.5) |
| ESN | 4096 | **78.1(.3)** | **80.0(.6)** | **88.5(.2)** | **92.6(.1)** | **83.0(.5)** | 87.9(1.0) | **86.1(.1)** | **83.1(.4)** | 73.4(.4) | **74.4(.3)** |
| InferSent-1 = paper, G | 4096 | 81.1 | 86.3 | 90.2 | 92.4 | 84.6 | 88.2 | 88.3 | 86.3 | 76.2 | 75.6 |
| InferSent-2 = fixed pad, F | 4096 | 79.7 | 84.2 | 89.4 | 92.7 | 84.3 | 90.8 | 88.8 | 86.3 | 76.0 | 78.4 |
| InferSent-3 = fixed pad, G | 4096 | 79.7 | 83.4 | 88.9 | 92.6 | 83.5 | 90.8 | 88.5 | 84.1 | 76.4 | 77.3 |
| Δ InferSent-3, BestRand | - | *1.6* | *3.4* | *0.4* | *0.0* | *0.5* | *2.0* | *2.4* | *1.0* | *2.3* | *2.9* |
| ST-LN | 4800 | 79.4 | 83.1 | 89.3 | 93.7 | 82.9 | 88.4 | 85.8 | 79.5 | 73.2 | 68.9 |
| Δ ST-LN, BestRand | - | *1.3* | *3.1* | *0.8* | *1.1* | *-0.1* | *0.5* | *-0.3* | *-3.6* | *-0.9* | *-5.5* |

Table 1: Performance (accuracy for all tasks except SICK-R and STSB, for which we report Pearson's $r$) on all ten downstream tasks where all models have 4096 dimensions with the exception of BOE (300) and ST-LN (4800). Standard deviations are show in parentheses. InferSent-1 is the paper version with GloVe (G) embeddings, InferSent-2 has fixed padding and uses FastText (F) embeddings, and InferSent-3 has fixed padding and uses GloVe embeddings. We also show the difference between the best random architecture (BestRand) and InferSent-3 and ST-LN, respectively. The average performance difference between the best random architecture and InferSent-3 and ST-LN is 1.7 and -0.4 respectively.

$$\mathbf{h} = \max(\text{ESN}(\mathbf{e}_1, \dots, \mathbf{e}_n)).$$

The property of echo state networks that sets them apart from randomly initialized classical recurrent networks, and allows for better performance, is the *echo state property*. The echo state property (Jaeger, 2001) claims that the state of the reservoir should be determined uniquely from the input history, and the effects of a given state asymptotically diminish in favor of more recent states.

In practice, one can satisfy the echo state property in most cases by ensuring that the spectral radius of $W^h$ is less than 1 (Lukoševičius & Jaeger, 2009). The spectral radius, i.e., the maximal absolute eigenvalue of $W^h$, is one of many hyperparameters to be tuned when using ESNs. Others include the activation function, the amount of leaking between states, the sparsity of $W^h$, whether to concatenate the inputs to the reservoir states, how to sample the values for $W^i$ and other factors. Lukoševičius & Jaeger (2009) gives a good overview of what hyperparameters are most critical when designing ESNs.

## 3.2 Evaluation

In our experiments, we evaluate on a standard sentence representation benchmark using SentEval (Conneau & Kiela, 2018). SentEval allows for evaluation on both downstream NLP datasets as well as probing tasks, which measure how accurately a representation can predict linguistic information about a given sentence. The set of downstream tasks we use for evaluation comprises sentiment analysis (MR, SST), question-type (TREC), product reviews (CR), subjectivity (SUBJ), opinion polarity (MPQA), paraphrasing (MRPC), entailment (SICK-E, SNLI) and semantic relatedness (SICK-R, STSB). The probing tasks consist of those in Conneau et al. (2018). We use the default SentEval settings (see Appendix A).

## 4 Results

We compare primarily to two well-studied sentence embedding models, InferSent (Conneau et al., 2017) and SkipThought (Kiros et al., 2015) with layer normalization (Ba et al., 2016). We point out that there are recently introduced multi-task sentence encoders that improve performance further, but these either do not use pre-trained word embeddings (GenSen (Subramanian et al., 2018)), or don't use SentEval (Universal Sentence Encoders (Cer et al., 2018)). Since both architectures are inspired by InferSent and SkipThought, and combine their respective supervised and unsupervised objectives, we limit our comparison to the original models.

We compute the average accuracy/Pearson's $r$, along with the standard deviation, over 5 different seeds for the random methods, and tune on validation for each task. See Appendix A for a discussion of the used hyperparameters.

Table 1 reports the results on the selected SentEval benchmark tasks, where all models have 4096 dimensions (with the exception of SkipThought, which has 4800). We compare to three different InferSent models: the results from the paper, which had non-standard pooling over padding symbols (InferSent-1); the results from the InferSent GitHub,[3] with fixed padding, using FastText instead of GloVe (InferSent-2); the results from an InferSent model we trained ourselves,[4] with fixed padding, using GloVe embeddings (InferSent-3) (see Appendix C for a more detailed discussion of padding and pooling).[5] Note that the comparison to layer-normalized SkipThought is not entirely fair, because it uses different (and older) word embeddings, but a higher dimensionality. We hypothesize that SkipThought might do a lot better if it had been trained with better pre-trained word embeddings.

First of all, we observe that all random sentence encoders generally improve over bag-of-embeddings. This is not entirely surprising, but it is important to note that the proper baseline for an $n$-dimensional sentence encoder is an $n$-dimensional BOREP representation, not an $(m < n)$-dimensional BOE representation. BOREP does markedly better than BOE, constituting a much stronger baseline (and requiring no additional computation besides a simple random projection).

When comparing the random sentence encoders, we observe that ESNs outperform BOREP and RandLSTM on all tasks. It is unclear whether (randomly initialized) LSTMs exhibit the echo state property, but the main reason for the improvement is likely that in our experiments ESNs had more hyperparameters available for tuning.

When comparing to InferSent, in which case we should look at InferSent-3 in particular (as it has fixed padding and also uses GloVe embeddings), we do see a clear difference on some of the tasks, showing that training does in fact help. The performance gains over the random methods, however, are not as big as we might have hoped, given that InferSent requires annotated data and takes time to train, while the random sentence encoders can be applied immediately. For SkipThought, we discern a similar pattern, where the gain over random methods (which do have better word embeddings) is even smaller. While SkipThought took a very long time to train, in the case of SICK-E you would actually even be better off simply using BOREP, while ESN outperforms SkipThought on five of the 10 tasks.

Note that in these experiments we do model selection over per-task validation set performance, but Appendix B shows that the method is robust, as we could also have used the best-overall model on validation and obtained similar results.

Keep in mind that the point of these results is not that random methods are better than these other encoders, but rather that we can get very close and sometimes even outperform those methods without any training at all, from just using the pre-trained word embeddings.

## 4.1 TAKING COVER TO THE MAX

If we take Cover's theorem to the limit, we can project to an even higher-dimensional representation as long as we can still easily fit things onto a modern GPU: hence, we project to $4096 \times 6$ (24576) dimensions instead of the 4096 dimensions we used in Table 1. In order to make for a fair comparison, we can also randomly project InferSent and SkipThought representations to the same dimensionality and examine performance.

Table 2 shows the results. Interestingly, the gap seems to get smaller, and the projection in fact appears to be detrimental to InferSent and SkipThought performance. The numbers reported in the table are competitive with (older) much more sophisticated trained methods.

---

[3] https://github.com/facebookresearch/InferSent

[4] We trained this model using the hyperparameters described by Conneau et al. (2017). Training on both SNLI (Bowman et al., 2015) and MultiNLI (Williams et al., 2018), we achieved a test performance on SNLI of 84.4 when max-pooling over padded words and 83.9 when max-pooling over the length of the sentences.

[5] We note that GenSen uses the same pooling as Infersent-1, and we show in Appendix C that this has a significant effect on performance.

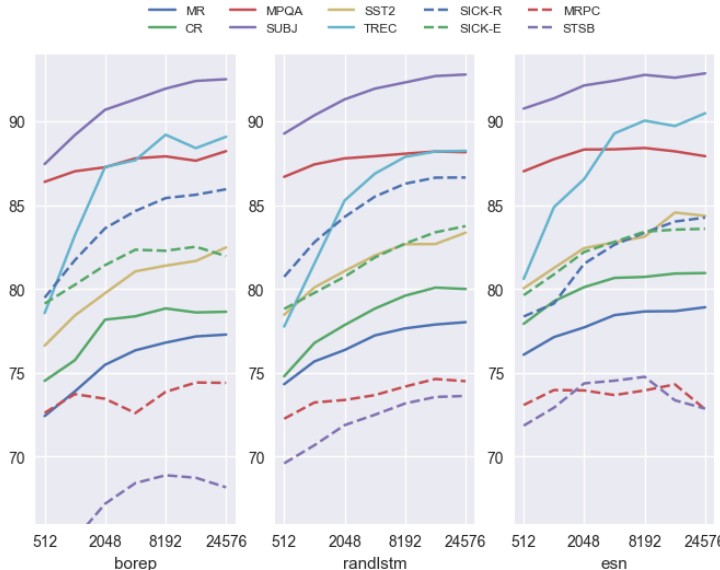

Figure 1: Performance while varying dimensionality, for the three random sentence encoders over all ten downstream tasks.

| Model | MR | CR | MPQA | SUBJ | SST2 | TREC | SICK-R | SICK-E | MRPC | STSB |
|---|---|---|---|---|---|---|---|---|---|---|
| BOE | 77.3(.2) | 78.6(.3) | 87.6(.1) | 91.3(.1) | 80.0(.5) | 81.5(.8) | 80.2(.1) | 78.7(.1) | 72.9(.3) | 70.5(.1) |
| BOREP | 78.6(.2) | 79.9(.4) | 88.8(.1) | 93.0(.1) | 82.5(.8) | 89.5(1.3) | 85.9(.0) | 84.3(.3) | 73.7(.9) | 68.3(.5) |
| RandLSTM | 78.2(.2) | 79.9(.4) | 88.2(.2) | 92.8(.2) | 83.2(.4) | 88.4(.7) | 86.6(.1) | 83.0(.9) | 74.7(.4) | **73.6(.4)** |
| ESN | **79.1(.2)** | **80.2(.3)** | **88.9(.1)** | **93.4(.2)** | **84.6(.5)** | **92.2(.8)** | **87.2(.1)** | **85.1(.2)** | **75.3(.6)** | 73.1(.2) |
| InferSent-3 4096×6 | **79.7** | **83.9** | **89.1** | **92.8** | **82.4** | **90.6** | 79.5 | **85.9** | **75.1** | **75.0** |
| ST-LN 4096×6 | 75.2 | 80.8 | 86.8 | 92.7 | 80.6 | 88.4 | **82.9** | 81.3 | 71.5 | 67.0 |

Table 2: Performance (accuracy for all tasks except SICK-R and STSB, for which we report Pearson's $r$) on all ten downstream tasks. Standard deviations are show in parentheses. All models have 4096×6 dimensions. ST-LN and InferSent-3 were projected to this dimension with a random projection.

Simply maximizing the number of dimensions, however, might lead to overfitting, so we also analyze how performance changes as a function of the dimensionality of the sentence embeddings: we sample random models for a range of dimensions, {512, 1024, 2048, 4096, 8192, 12288, 24576}, and train models for BOREP, random LSTMs, and ESNs. Performance of these models is shown in Figure 1.

As suggested by Cover's theorem, as well as earlier findings in the sentence embedding literature (see e.g. Fig. 5 of Conneau et al. (2017)), we observe that higher dimensionality in most cases leads to better performance. In some cases it looks like we would have benefited from having even higher dimensionality (e.g. SUBJ, TREC and SST2), while in other cases we can see that the model probably starts to overfit (STSB, SICK-E for BOREP). In general, the trend is up, meaning that a higher-dimensional embeddings leads to better performance.

## 5 ANALYSIS

We analyze random sentence embeddings by examining how these embeddings perform on the probing tasks introduced by Conneau et al. (2018), in order to gauge what properties of sentences they are able to recover. These probing tasks were introduced in order to provide a framework for ascertaining the linguistic properties of sentence embeddings, comprising three types of information: surface, syntactic and semantic information.

| Model | SentLen | WC | TreeDepth | TopConst | BShift | Tense | SubjNum | ObjNum | SOMO | CoordInv |
|---|---|---|---|---|---|---|---|---|---|---|
| BOE (300d, class.) | 60.5 | 87.5 | 32.0 | 62.7 | 50.0 | 83.7 | 78.0 | 76.6 | 50.5 | 53.8 |
| BOREP (4096d, class.) | 64.4 | **97.1** | 33.0 | 71.3 | 49.8 | 86.3 | 81.5 | 79.3 | 49.5 | 54.1 |
| RandLSTM (4096d, class.) | 72.8 | 94.1 | 35.6 | 76.2 | 55.2 | 86.6 | 84.0 | 79.5 | 49.7 | 63.1 |
| ESN (4096d, class.) | 78.8 | 92.4 | 36.9 | 76.2 | 62.9 | 86.6 | 82.3 | 79.7 | 49.7 | 60.3 |
| Infersent-3 | **80.6** | 93.5 | 37.1 | 78.2 | 57.3 | 86.8 | 84.8 | 80.5 | 53.0 | 65.8 |
| ST-LN | 79.9 | 79.9 | **39.5** | **82.1** | **69.4** | **90.2** | **86.2** | **83.4** | **54.5** | **68.9** |

Table 3: Performance on a set of probing tasks defined in (Conneau et al., 2018). All random architecture models are 4096 dimensions and were selected by tuning over validation performance on the classification tasks.

There are two surface information tasks: predicting the correct length bin from 6 equal-width bins sorted by sentence length (Length) and predicting which word is present in the given sentence from a set of 1000 mid-frequency words (Word Content, WC). Syntactic information comprises 3 tasks: predicting whether a sentence has been perturbed by switching two adjacent words (BShift); the depth of the constituent parse tree of the sentence (TreeDepth); and the topmost constituent sequence of the constituent parse in a 20-way classification problem (TopConst). Finally, there are five semantic information tasks: predicting the tense of the main-clause verb (Tense); the number of the subject of the main clause (SubjNum); the number of the direct object of the main clause (ObjNum); whether a sentence has been modified by replacing a noun or verb with another in a way that the newly formed bigrams have similar frequencies to those they replaced (Semantic Odd Man Out, SOMO); and whether the order of two coordinate clauses has been switched (CoordInv).

Table 3 shows the performance of the random sentence encoders (using the best-overall model tuned on the classification validation sets of the SentEval tasks) on these probing tasks along with bag-of-embeddings (BOE), SkipThought-LN, and InferSent. From the table, we see that ESNs and RandL-STMs outperform BOE and BOREP on most of the tasks, especially those that require knowledge of the order of the words. This indicates that these models, even though initialized randomly, are capturing order, as one would expect. We also see that ESNs and InferSent are fairly close on many of the tasks, with Skipthought-LN generally outperforming both.

It seems that random models do best when the tasks largely require picking up on certain words. We can see which tasks these are by looking at how well BOREP does compared to the recurrent models (WC, Tense, SubjNum, ObjNum are good candidates for this type of task). In these, random models are all very competitive to the trained encoders. If one looks at the tasks where there is the largest difference between ESN and the best of IS or ST-LN (SOMO, CoordInv, BShift, TopConst) it seems that one thing these all have in common is that they do require sequential knowledge. We say this because the BOREP baseline lags behind the recurrent models significantly for many of these (and is often at the majority-vote baseline) and also because of the very definitions of these tasks. This also makes intuitive sense as well since this is the type of knowledge that is much harder to learn and is not provided by stand-alone word embeddings. Therefore, we'd expect the trained models to have an edge here, which seems to bear out in these experiments.

## 6 DISCUSSION

In light of our findings, we list several take-away messages with regard to sentence embeddings:

- If you need a baseline for your sentence encoder, don't just use BOE, use BOREP of the same dimension, and/or a randomly initialized version of your encoder.

- If you are pressed for time and have a small to mid-size dataset, simply randomly project to a very high dimensionality, and profit.

- More dimensions in the encoder is usually better (up to a point).

- If you want to show that your system is better than another system, use the same classifier on top with the same hyperparameters; and use the same word embeddings at the bottom; while having the same sentence embedding dimensionality.

- Be careful with padding, pooling and sorting: you may inadvertently end up favoring certain methods on some tasks, making it harder to identify sources of improvement.

As Rahimi and Recht wrote when reflecting on their random kitchen sinks paper[6]:

> Its such an easy thing to try. When they work and I'm feeling good about life, I say "wow, random features are so powerful! They solved this problem!" Or if I'm in a more somber mood, I say "that problem was trivial. Even random features cracked it." [...] Regardless, it's an easy trick to try.

Indeed, random sentence encoders are easy to try: they require no training, and should be used as a solid baseline to be compared against when learning sentence encoders that are supposed to capture more than simply what is encoded in the pre-trained word embeddings. While sentence embeddings constitute a very promising research direction, much of their power appears to come from pre-trained word embeddings, which even random methods can exploit. The probing analysis revealed that the trained systems are in fact better at some more intricate semantic probing tasks, aspects of which are however apparently not well-reflected in the downstream evaluation tasks.

## 7 CONCLUSION

In this work we have sought to put sentence embeddings on more solid footing by examining how much trained sentence encoders improve over random sentence encoders. As it turns out, differences exist, but are smaller than we would have hoped: in comparison to sentence encoders such as SkipThought (which was trained for a very long time) and InferSent (which requires large amounts of annotated data), performance improvements are less than 2 points on average over the 10 SentEval tasks. Therefore one may wonder to what extent sentence encoders are worth the attention they're receiving. Hope remains, however, if we as a community start focusing on more sophisticated tasks that require more sophisticated learned representations that cannot merely rely on having good pre-trained word embeddings.

ACKNOWLEDGMENTS

We would like to thank Alexis Conneau, Kyunghyun Cho, Jason Weston, Maximilian Nickel and Armand Joulin for useful feedback and suggestions.

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

## A    HYPERPARAMETERS

For all experiments, we attempt to keep the number of tunable hyperparameters to a minimum. By being judicious with the number of tuning experiments and averaging over different seeds, we provide strong evidence that these architectures are robust and can be competitive with trained (non-random) sentence embedding models.

In all experiments, we tune the type of pooling to use. Different tasks benefit from different types of pooling, and while many pooling mechanisms have been proposed in the literature, we just tune

over the most commonly used ones: mean pooling and max pooling. We use the publicly available 300-dimensional GloVe embeddings (Pennington et al., 2014) trained on Common Crawl for all experiments. All words that are not in the vocabulary for GloVe are assigned a vector of zeros.

For the ESNs, we only tune whether to use a ReLU or no activation function,[7] the spectral radius from $\{0.4, 0.6, 0.8, 1.0\}$, the range of the uniform distribution for initializing $W^i$ where the max distance from zero is selected from $\{0.01, 0.05, 0.1, 0.2\}$, and finally the fraction of elements in $W^h$ that are set to 0, i.e., sparsity, is selected from $\{0, 0.25, 0.5, 0.75\}$. Furthermore, our model did not include a bias term $b^i$.

We chose not to experiment with other possibilities that ESNs provide that could further enhance performance like leaking or concatenating/adding the input embedding to the reservoir state in favor of a simpler model.

We use the default SentEval settings, which are to train with a logistic regression classifier, use a batch size of 64, a maximum number of epochs of 200 with early stopping,[8] no dropout, and use Adam (Kingma & Ba, 2014) for optimization with a learning rate of 0.001.

# B  TESTING ROBUSTNESS

| Model | MR | CR | MPQA | SUBJ | SST2 | TREC | SICK-R | SICK-E | MRPC | STSB |
|---|---|---|---|---|---|---|---|---|---|---|
| BOREP (class.) | 77.6(.4) | 79.3(.2) | **88.3(.2)** | 91.9(.2) | 80.6(.6) | **88.8(.3)** | 85.5(.1) | 82.2(.1) | 73.6(.8) | 69.3(.6) |
| BOREP (corr.) | 77.4(.4) | 79.6(.4) | **88.3(.1)** | 92.2(.1) | 81.8(.4) | 85.6(1.4) | 84.6(.2) | 82.1(.3) | 73.9(.4) | 68.5(.6) |
| RandLSTM (class., corr.) | 77.2(.3) | 78.7(.5) | 87.9(.1) | 91.9(.2) | 81.5(.3) | 86.5(1.1) | 85.5(.1) | 81.8(.5) | **74.1(.5)** | 72.4(.5) |
| ESN (class.) | **78.2(.2)** | **80.1(.2)** | **88.3(.2)** | **92.5(.1)** | **83.0(.5)** | 88.3(1.5) | 85.3(.2) | 82.4(.7) | 73.1(.4) | 70.0(.4) |
| ESN (corr.) | 76.7(.2) | 78.2(.6) | 88.0(.1) | 91.5(.3) | 81.2(.5) | 86.7(1.2) | **86.1(.1)** | **82.9(.1)** | **74.1(.5)** | **74.4(.3)** |

Table 4: Performance (accuracy for all tasks except SICK-R and STSB, for which we report Pearson's $r$) on all ten downstream tasks. Standard deviations are show in parentheses. All models have 4096 dimensions and were selected by tuning over validation performance on classification tasks or correlation tasks as noted. For RandLSTM this corresponds to a single model that uses max pooling.

In order to examine the stability of the random sentence encoders, we select the two best overall models by best validation score—one that achieved the highest accuracy score, and one that achieved the highest correlation score (as these differed significantly)—and examine the results. The performance of these models is shown in Table 4. We observe that performance is very stable, and that task-specific tuning yields little or no benefit over the best-overall model, which is beneficial: the good results obtained by random sentence encoders are not some fluke, and the finding is robust.

# C  POOLING AND PADDING

| | Model | MR | CR | MPQA | SUBJ | SST2 |
|---|---|---|---|---|---|---|
| Sorted | RandLSTM | 81.7 | 84.0 | 89.4 | 93.0 | 81.2 |
| | InferSent | 81.6 | 86.7 | 90.3 | 92.5 | **84.5** |
| | GenSen | **82.7** | **87.4** | **91.0** | **94.1** | 83.2 |
| Unsorted | RandLSTM | 77.2 | 79.2 | 88.1 | 92.0 | 81.8 |
| | InferSent | **79.9** | **84.3** | 89.5 | **92.4** | **84.4** |
| | GenSen | 78.1 | 84.2 | **89.7** | **92.4** | 83.9 |

Table 5: Accuracy on single-sentence binary classification tasks from SentEval, where max-pooling is done over padded values instead of over the length of the sentence. Experiments are split between *Sorted* where sentences are sorted in order of length prior to batching and *Unsorted* where they are not.

---

[7] A tanh activation did not work well in these experiments, even though it is often used in ESNs.

[8] Training is stopped when validation performance has not increased 5 times. Checks for validation performance occur every 4 epochs.

We further analyzed how max-pooling over padding affects downstream evaluations and noticed that for this effect to occur, the batch size to produce the embeddings and the order in which sentenced were embedded needed to be a specific way. The order in which sentences are embedded in SentEval is not random, as sentences are sorted by length prior to being grouped into batches. We noticed that upsetting this order, or changing the batch size so that sentences are grouped differently, causes a significant change on the downstream performance.

In Table 5, we reproduce this effect for RandLSTM (averaged over 5 seeds) and also include results for InferSent and GenSen using their released code. The first half of the table shows results when max pooling with padding is used and the batches are sorted. The second half of the table shows performance when the batches are unsorted. As can be seen by the table, the performance has a significant drop-off when the batches are unsorted, especially for MR and CR.

Max pooling over padded values changes negative values in the features of longer sentences to zero. This is because if the largest value in the hidden representations over the length of the sentence is negative, the padded zeros will be greater. Thus, longer sentences, when grouped with shorter ones, will have more sparse representations. We tried to reproduce this effect by using a ReLU, but it didn't increase performance. We also checked to see if length was strongly correlated with either class for the problems in Table 5, but found the correlation was low for all binary tasks. In fact it is 0.0 for CR, one of the tasks most affected by this phenomenon.

It turns out that in several of the datasets, those examples obtaining the sparse representations (becoming *sparsed*) occur much more often in certain classes than others. Table 6 analyzes the distribution of sparsed examples in MR, CR, MPQA, SUBJ, and SST2.

Since MR, CR, MPQA, and SUBJ have no defined training/validation/testing split, the first row of the table shows the percentage of the data that becomes sparsed when the sentences are embedded. The second row is what percentage of those sparsed embeddings are of the positive class.

SST2 is split into a training/validation/test set. In it, only 3.5% of the training data is sparsed. However, 94% of the sparsed data has a class 1 label. In the testing data, 75% of the data is sparsed with 61% of the data having class 1. Overall, since little training data is actually sparsed, and the class imbalance of the testing data isn't as skewed as in the other datasets, SST2 is less affected by these sparsed representations as can be seen in the table.

|                       | MR | CR | MPQA | SUBJ |
|-----------------------|----|----|------|------|
| % Total data sparsed  | 23 | 59 | 8    | 23   |
| % Class 1 data sparsed| 91 | 82 | 70   | 92   |

Table 6: Percentage of total data sparsed due to max-pooling over padded values and the percentage of that data that is the positive class for MR, CR, MPQA, and SUBJ.

## D EXPLORING DIFFERENT INITIALIZATION STRATEGIES

We analyzed the effects different initialization strategies had on performance for BOREP and RandLSTM. We experimented with 6 different initialization strategies: 1) Heuristic, which is the approach used in this paper for BOREP and RandLSTM experiments unless otherwise noted, where elements are sampled uniformly from $[-\frac{1}{\sqrt{d}}, \frac{1}{\sqrt{d}}]$ 2) Uniform where parameters are sampled from $[-0.1, 0.1]$, 3) Normal where parameters are sampled from a normal distribution with 0 mean and a standard deviation of 1, 4) Orthogonal where elements are sampled from the heuristic and then the matrices of the parameters are made orthogonal 5) He initialization (He et al., 2015) and 6) Xavier initialization (Glorot & Bengio, 2010).

We find that BROEP is robust to initialization strategy, with a slight edge for Orthogonal and Xavier initialization. However, RandLSTM does poorly with Normal initialization and to a lesser degree, Uniform. He and Xavier initialization seem to give the best average performance.

| | Model | MR | CR | MPQA | SUBJ | SST2 | TREC | SICK-R | SICK-E | MRPC | STSB | Avg. |
|---|---|---|---|---|---|---|---|---|---|---|---|---|
| BOREP | Heuristic | 77.4(.4) | 79.5(.2) | 88.3(.2) | 91.9(.2) | 81.8(.4) | **88.8(.3)** | 85.4(.2) | 82.2(.1) | 73.9(.4) | 72.1(.5) | 81.8 |
| | Uniform | **77.7(.2)** | **80.2(.2)** | 88.4(.2) | **92.2(.2)** | 81.7(.2) | 87.6(.7) | 85.4(.2) | 82.2(.1) | 73.7(.4) | 72.1(.5) | 81.6 |
| | Normal | **77.7(.2)** | 79.6(.4) | 88.4(.1) | **92.2(.1)** | 81.0(.4) | 88.6(1.4) | 85.1(.1) | 82.1(.6) | 73.3(.7) | 72.1(.5) | 81.5 |
| | Orthogonal | 77.6(.1) | 79.3(.4) | **88.5(.2)** | 92.0(.1) | 81.0(.4) | 87.2(.9) | **85.6(.1)** | **82.5(.2)** | **74.1(.3)** | **73.0(.6)** | 82.1 |
| | He | 77.5(.2) | 79.7(.3) | 88.4(.2) | **92.2(.2)** | 81.6(.3) | 88.3(.7) | 85.4(.2) | 82.0(.3) | 73.5(.6) | 72.1(.5) | 81.6 |
| | Xavier | 77.6(.1) | 79.6(.2) | 88.4(.1) | 92.1(.1) | **81.9(.3)** | 86.8(.7) | 85.4(.2) | **82.5(.1)** | 74.0(.3) | 72.1(.5) | 82.0 |
| RandLSTM | Heuristic | 77.2(.3) | 78.7(.5) | 87.9(.1) | 91.9(.2) | 81.5(.3) | 86.5(1.1) | **85.5(.1)** | 81.8(.5) | 74.1(.5) | 72.4(.5) | 81.8 |
| | Uniform | 76.4(.3) | 78.6(.6) | 87.9(.2) | 91.6(.2) | 81.0(.5) | 88.7(.8) | 82.9(.2) | 81.1(.2) | 73.3(1.2) | 66.7(.8) | 80.8 |
| | Normal | 67.0(.4) | 69.8(.3) | 85.8(.3) | 84.2(.2) | 71.9(1.3) | 85.0(1.2) | 58.8(.9) | 69.2(.4) | 68.4(.6) | 33.9(.9) | 69.4 |
| | Orthogonal | 77.1(.1) | 78.4(.1) | 87.8(.2) | 91.5(.3) | 81.7(.5) | 85.5(1.7) | 85.5(.2) | 82.0(.3) | 74.3(.3) | 72.5(.5) | 81.6 |
| | Kaiming | **77.9(.2)** | **79.7(.2)** | **88.3(.1)** | **92.5(.2)** | **82.9(.6)** | **88.9(1.0)** | 85.0(.1) | **83.5(.5)** | **74.8(.3)** | 69.8(.5) | **82.3** |
| | Xavier | 77.3(.1) | 78.9(.4) | 88.0(.2) | 91.9(.1) | 81.7(.3) | 86.8(1.3) | 85.3(.1) | 81.6(1.1) | 74.1(.2) | **72.7(.3)** | 81.8 |

Table 7: Performance (accuracy for all tasks except SICK-R and STSB, for which we report Pearson's $r$) on all ten downstream tasks. Standard deviations are show in parentheses. Six different approaches to initializing the parameters for each model are explored.

| | Model | MR | CR | MPQA | SUBJ | SST2 | TREC | SICK-R | SICK-E | MRPC | STSB |
|---|---|---|---|---|---|---|---|---|---|---|---|
| BOE (300d) | Heuristic | **62.9(.5)** | **72.0(.7)** | 73.2(.6) | **80.5(.2)** | 61.8(.6) | **72.4(1.9)** | 70.9(.4) | 76.6(.2) | 70.5(1.3) | 54.7(.7) |
| | Uniform | 61.0(.5) | 67.5(.5) | 73.6(.5) | 78.6(.1) | **62.4(1.0)** | 68.1(1.0) | 67.7(.8) | 74.9(.8) | 69.3(.6) | 59.3(.8) |
| | Normal | 62.4(.5) | 69.7(.8) | 73.2(.3) | 79.7(.3) | 62.2(.4) | 70.6(2.4) | **72.5(.7)** | **77.8(.6)** | **71.7(.5)** | **63.4(.5)** |
| | Orthogonal | 59.4(.5) | 63.8(.0) | 72.4(.4) | 74.9(.8) | 61.8(.7) | 59.2(2.3) | 72.3(.4) | 75.6(1.4) | 66.5(.0) | 63.1(.7) |
| | He | 61.3(.5) | 69.1(.5) | **73.7(.5)** | 78.9(.3) | 62.1(1.3) | 68.5(1.0) | 67.8(.7) | 75.3(.5) | 68.8(1.3) | 59.1(.8) |
| | Xavier | 58.5(.3) | 63.8(.0) | 72.9(.3) | 74.6(.5) | 61.9(1.3) | 61.2(1.4) | 67.7(.8) | 71.1(1.3) | 66.5(.0) | 59.4(.9) |
| BOE (4096d) | Heuristic | 71.8(.3) | **78.2(.2)** | 81.5(.4) | **88.5(.1)** | 74.5(1.1) | **85.0(1.3)** | 76.9(4.2) | 81.5(.2) | 71.8(.9) | 57.0(.2) |
| | Uniform | 71.8(.4) | 77.0(.5) | 83.2(.2) | 88.0(.1) | 75.8(.4) | 84.8(.5) | 78.4(.3) | 80.3(.4) | 71.7(.4) | 64.6(1.5) |
| | Normal | **72.1(.4)** | 77.7(.2) | 82.9(.3) | 88.2(.3) | 75.6(.5) | **85.0(1.2)** | 81.0(.2) | 82.3(.3) | **73.6(.5)** | 66.4(.8) |
| | Orthogonal | 67.6(.3) | 75.2(.5) | **83.3(.3)** | 85.0(.3) | 75.3(.5) | 78.2(4.2) | 81.0(.3) | 81.6(.3) | 72.8(1.2) | **68.4(.5)** |
| | He | 70.4(.7) | 76.0(.2) | 82.7(.3) | 87.1(.3) | **76.0(.4)** | 83.0(1.1) | 78.4(.2) | 79.2(.4) | 70.8(.6) | 65.6(1.4) |
| | Xavier | 68.7(.4) | 75.3(.5) | 82.3(.3) | 85.4(.4) | 75.2(.4) | 80.8(1.6) | 78.4(.2) | 78.9(.5) | 70.4(.2) | 66.3(1.3) |
| BOREP | Heuristic | **69.3(.3)** | **75.6(.3)** | 80.4(.5) | 86.5(.4) | 72.5(.5) | 81.0(.9) | 78.8(.4) | 81.3(.2) | 72.2(1.0) | 57.7(.5) |
| | Uniform | 68.2(.3) | 74.0(.4) | 79.7(.3) | 85.1(.4) | 72.9(.6) | 81.0(2.3) | 80.9(.1) | **82.1(.2)** | 73.3(.9) | 67.2(.7) |
| | Normal | 68.8(.6) | 75.4(.5) | 80.2(.7) | 86.4(.2) | 72.9(.8) | **83.2(.9)** | 80.6(.3) | 81.5(.3) | 72.9(1.0) | 63.0(.9) |
| | Orthogonal | 61.6(.2) | 65.3(.4) | 75.4(.4) | 80.1(.4) | 69.4(.8) | 67.6(1.2) | **81.0(.3)** | 78.4(.4) | 71.6(.8) | 68.0(.4) |
| | He | 68.7(.5) | 74.7(.8) | 80.3(.2) | 86.2(.4) | **73.6(.5)** | **83.2(1.2)** | 80.8(.1) | **82.1(.3)** | **73.9(.9)** | 65.3(.8) |
| | Xavier | 62.8(.5) | 69.3(1.0) | 76.4(.5) | 80.9(.6) | 71.0(.8) | 67.7(1.6) | 80.9(.1) | 79.4(.8) | 72.4(.7) | **67.9(.4)** |
| RandLSTM | Heuristic | **66.4(.5)** | **73.5(.2)** | 76.7(.2) | **84.6(.3)** | **70.3(.6)** | 79.8(1.0) | **78.8(.3)** | **82.1(.3)** | 71.2(1.0) | 62.4(.7) |
| | Uniform | 59.6(.6) | 68.0(0.6) | 74.7(.4) | 76.5(.7) | 62.5(1.1) | 78.0(2.6) | 73.7(.3) | 78.5(.4) | **71.3(.4)** | 55.9(.4) |
| | Normal | 53.2(.7) | 62.6(.5) | **77.5(.2)** | 64.2(.9) | 54.2(1.3) | 78.9(1.1) | 53.9(1.0) | 68.0(.5) | 66.6(.7) | 29.5(3.2) |
| | Orthogonal | 54.2(1.3) | 63.8(.0) | 69.1(.2) | 67.2(1.2) | 55.9(2.2) | 53.8(2.1) | 74.5(.4) | 69.1(.5) | 66.5(.1) | 59.3(.7) |
| | Kaiming | 63.9(.6) | 72.8(.4) | 75.3(.7) | 83.3(.4) | 68.9(.7) | **80.2(2.9)** | 77.9(.3) | 81.0(.4) | 71.0(.6) | 62.1(.4) |
| | Xavier | 57.0(.1) | 63.8(.0) | 70.5(.5) | 69.7(.8) | 61.7(1.3) | 61.4(3.8) | 77.2(.4) | 76.1(.8) | 66.7(.4) | **62.6(.4)** |
| Δ GloVe, Random | | *8.6* | *4.6* | *8.1* | *6* | *9.3* | *5.7* | *4.6* | *1.4* | *0.9* | *5.1* |

Table 8: Performance (accuracy for all tasks except SICK-R and STSB, for which we report Pearson's $r$) on all ten downstream tasks. Standard deviations are show in parentheses. All parameters for both the underlying word embeddings and architectures (if applicable) are randomly sampled from six different initialization schemes. The last row shows the performance difference between the best performing model (from BOREP and RandLSTM) in Table 7 which uses pre-trained word embeddings and the best performing model from this table (outside of BOE with 4096 dimensino vectors) which uses randomly initialized embeddings. The average gain from using pre-trained embeddings is 5.4 points.

# E   EXPLORING RANDOM ENCODERS WITH RANDOM WORD EMBEDDINGS

We next explore the contributions of pre-trained word embeddings for the BOREP and RandLSTM models. In these experiments, both the parameters of the architectures and the word embeddings are randomly sampled. Just as in Section D, we experiment with six ways of initializing the random embeddings for BOE, BOREP, RandLSTM, and a 4096 dimension BOE model. The parameters of the architectures (if applicable) are sampled in the same way as the random embeddings. The results are shown in Table 8.

From the table, we see knowledge from the pre-trained word embeddings offers significant improvement for most tasks, averaging 5.4 points per task. It seems to be especially helpful for classification,

and less-so for those tasks relying on a measure of semantic similarity. This is sensible because semantic similarity tasks can make use of embeddings appearing in the test data that were not in the training data if the words appear in both sentences in the example. However, this is not the case for classification as an unseen random word embeddings provide little information.

Interestingly, pooling randomly initialized 4096 dimension embeddings outperforms BOREP and RandLSTM. It would be interesting to see the performance of this model when the embeddings are pre-trained instead of random. Perhaps pooling them would be competitive with trained encoders like Infersent. It would also be interesting to see what effect large-dimension word embeddings have on both a trained sentence encoder like Infersent as well as random recurrent architectures. We leave this exploration for future work.

