# OpenReview forum: "No Training Required: Exploring Random Encoders for Sentence Classification"
_ICLR.cc/2019/Conference_

### Official Review · AnonReviewer2 · 2018-11-03
**Strong, clear paper with worthwhile contribution**

**Rating:** 8
**Confidence:** 4

**Review:**

This paper tests a number of untrained sentence representation models - based on random embedding projections, randomly-initialized LSTMs, and echo state networks - and compares the outputs of these models against influential trained sentence encoders (SkipThought, InferSent) on transfer and probing tasks. The paper finds that using the trained encoders yields only marginal improvement over the fully untrained models.

I think this is a strong paper, with a valuable contribution. The paper sheds important light on weaknesses of current methods of sentence encoding, as well as weaknesses of the standard evaluations used for sentence representation models - specifically, on currently-available metrics, most of the performance achievements observed in sentence encoders can apparently be accomplished without any encoder training at all, casting doubt on the capacity of these encoders - or existing downstream tasks - to tap into meaningful information about language. The paper establishes stronger and more appropriate baselines for sentence encoders, which I believe will be valuable for assessment of sentence representation models moving forward.

The paper is clearly written and well-organized, and to my knowledge the contribution is novel. I appreciate the care that has been taken to implement fair and well-controlled comparisons between models. Overall, I am happy with this paper, and I would like to see it accepted.

Additional comments:

-A useful addition to the reported results would be confidence intervals of some kind, to get a sense of the extent to which the small improvements for the trained encoders are statistically significant.

-I wonder about how the embedding projection method would compare to simply training higher-dimensional word embeddings from the start. Do we expect substantial differences between these two options?

---

> ### Author Response · Authors · 2018-11-26
> **Thank you for your feedback and review!**
>
> Thank you for your feedback and review! We have added standard deviations for the experiments.
>
> Trying high dimensional word embeddings is a very interesting idea. We did not have the time to implement this before the rebuttal deadline (we would like them to be trained on near the same amount of data as the released GloVe embeddings are), but do plan to try this out as soon as we can. Thank you for this idea! In Appendix E, we did experiment (and compare with BOREP and RandLSTM) with pooling 4096 dimensional random word embeddings. They seem to outperform the other models for the same dimensionality, which provides some evidence that large, trained embeddings could achieve strong performance. It would be interesting to see if BOREP, RandLSTM, and ESN improve as well with using large embeddings.

---

### Official Review · AnonReviewer1 · 2018-11-08
**Interesting results though lacks thorough analysis**

**Rating:** 7
**Confidence:** 4

**Review:**

This paper proposes that randomly encoding a sentence using a set of pretrained word embeddings is almost as good as using a trained encoder with the same embeddings. This is shown through a variety of tasks where certain tasks perform well with a random encoder and certain ones don't.

The paper is well written and easy to understand and the experiments show interesting findings. There is a good analysis on how the size of the random encoder affects performance which is well motivated by Cover's theorem.

However, the random encoders that are tested in the paper are relatively limited to random projections of the embeddings, a randomly initialized LSTM and an echo state network. Other comparisons would make the results significantly more interesting and would move away from the big assumption stated in the first sentence, i.e. that sentence embeddings are: "learned non-linear recurrent combinations". Some major models that are missed by this include paragraph vectors (which do not require any initial training if initialized with pretrained word embeddings), CNNs and Transformers. Given this, the takeaways from this paper seem quite limited to recurrent representations and it's unclear how it would generalize to other representations.

An additional problem is that the paper states that ST-LN used different and older word embeddings which may make the comparison flawed when compared with the random encoders. In this case, the only fairly trained sentence encoder that is compared with is InferSent. The RandLSTM also has an issue in that the biases are intialized around zero whereas it's well known that using an initially higher forget gate bias significantly improves the performance of the LSTM.

Finally, the analysis of the results seems weak. The tasks are very different from each other and no reason or potential explanation is given why certain tasks are better than others with random encoders, except for SOMO and CoordInv. E.g. Could some tasks be solved by looking at keywords or bigrams? Do some tasks intrinsically require longer term dependencies? Do some tasks have more data?

Other comments:
- The results and especially random encoder results should be shown with confidence intervals.
- Section 3.1.3 the text refers to W^r but that does not appear in any equations.

=== After rebuttal ===
Thanks for adding the additional experiments (particularly with fully random embeddings) and result analyses to the paper. I feel that this makes the paper stronger and have raised my score accordingly.

---

> ### Author Response · Authors · 2018-11-26
> **Thank you for your review and comments! (1/2)**
>
> Thank you for your review and comments! We have incorporated your feedback into our latest draft.
>
> We focused on recurrent architectures in this paper because that is the type of network used by the top performing models within in this evaluation framework. Therefore, by using a recurrent model, we capture the prior of these state-of-the-art models and this gives us a better understanding of how much these published approaches benefit from learning. Models like InferSent (Conneau et al. 2017), GenSen (Subramanian et al. 2018), SkipThought (Kiros et al. 2015), Dissent (Nie et al. 2017), Byte mLSTM (Radford et al. 2017), all use recurrent models. While there are some architectures in the literature that use CNNs (like Gan et al. (2016)) they are not among the current state-of-the-art. The point of this work is to provide baselines - which means CNNs and Transformers can be compared to our numbers, and should hopefully be able to beat them.
>
> Another attractive reason for using recurrent networks is that they have very few hyperparameters to tune, in fact the only hyperparameter we varied was the hidden size in our experiments (and we detailed what size this was in our results). Architectures like CNNs or transformers require more design decisions and do not have a "default architecture"  which leads to a lot more experimentation and tuning.
>
> Regarding using paragraph vector, it actually is a trained model and the results on these downstream tasks are not very competitive (see Hill et al. 2016 for the numbers). We'd be happy to include it in our results, but we don't think it would add to the message of the paper.
>
> We do agree that sentence embeddings are more general than "learned non-linear recurrent combinations" and have changed this in the current iteration of the paper. Thanks for pointing this out!
>
> We also agree that the comparison of ST-LN isn't quite as even as we would like, which is why we did make that note in our original submission that ST-LN could potentially be higher if they used GloVe embeddings. The problem with making this comparison is simply that reproducing ST-LN takes about a month of computation time. However, others have experimented with ST-LN with Glove embeddings. Results for this model are in https://arxiv.org/pdf/1707.06320.pdf for example, with an older evaluation setup more comparable to numbers in http://aclweb.org/anthology/Q16-1002. GenSen also experiments with a SkipThought model, and while not initialized with GloVe, they project from GloVe into their embedding space. They actually found this to work better than just using GloVe in their experiments (confirmed through correspondence with the authors). We do compare to the full GenSen model in the appendix, and their version that has just ST is included in their paper. It was one of their baselines which was handily beaten by their full model. So while a direct comparison is tricky, we can safely say that adding GloVe to ST-LN would not elevate the model to a level that would change the message of this paper.

---

> > ### Author Response · Authors · 2018-11-26
> > **Thank you for your review and comments! (2/2)**
> >
> > Regarding initializing the biases, we don't see the weakness in initializing our biases as we do - we use the standard initialization procedure. If we initialize the forget gate bias as you suggest (from Jozefowicz et al. 2015), we might even get better results. However, the reason for initializing the biases in this way is because of gradient flow during optimization. Since we're not doing any training, it's not that relevant here.
> >
> > As per your suggestion, we did update the analysis in the new version, thank you. All of these tasks have the same amount of training/validation/testing data and are balanced so the amount of training data has no effect on performance. It seems that random models do best for tasks requiring picking up on certain words: we can see which tasks these are by looking at how well BOREP does compared to the recurrent models (so WC, Tense, SubjNum, ObjNum are good candidates for this type of task). In these tasks, random models are all very competitive to the trained encoders. If one looks at the tasks where there is the largest difference between ESN and max(IS/ST-LN), which are SOMO, CoordInv, BShift, TopConst, it seems that these all have in common that they do require some sequential knowledge. We say this because the BOREP baseline lags behind the recurrent models significantly for many of these (especially when considering where the majority-vote baseline is) and it also makes sense that this is the case when one looks at the definitions of these tasks. This also makes intuitive sense, as this type of knowledge is much harder to learn and is not provided by just pure word embeddings, and so we'd expect the trained models to have an edge here, which seems to bear out in these experiments. We also added further analysis of various other questions in the appendix. We hope you find the updated and more detailed analysis more to your liking.
> >
> > We have added confidence intervals in our latest version, and we changed W^r to W^h. Thanks for pointing these out.

---

### Official Review · AnonReviewer3 · 2018-11-11
**interesting investigation with worthwhile contribution; some suggested areas of improvement**

**Rating:** 7
**Confidence:** 4

**Review:**

This paper is about exploring better baselines for sentence-vector representations through randomly initialized/untrained networks. I applaud the overall message of this paper that we need to evaluate our models more thoroughly and have better baselines. The experimentation is quite thorough and I like that you
1) explored several different architectures
2) varied the dimensionality of representations
3) examine representations with probing tasks in the Analysis section.

Main Critique
- In your takeaways you say that, “For some of the benchmark datasets, differences between random and trained encoders are so small that it would probably be best not to use those tasks anymore.” I don’t think this follows from your results. Just because current trained encoders do not perform better than random encoders on these tasks doesn’t in itself mean these tasks aren’t good evaluation tasks. These tasks could be faulty for other reasons, but just because we have no better technique than random encoders currently, doesn’t make these evaluation tasks not worthwhile. Perhaps you could further examine what features (n-gram, etc.) it takes to do well on these tasks in order to argue that they shouldn’t be used.
- In your related work section you say that “We show that a lot of information may be crammed into vectors using randomly parameterized combinations of pre-trained word embeddings: that is, most of the power in modern NLP systems is derived from having high-quality word embeddings, rather than from having better encoders.” Did you run experiments with randomly initialized embeddings? This paper (https://openreview.net/forum?id=ryeNPi0qKX) finds that representations from LSTMs with randomly initialized embeddings can perform quite well on some transfer tasks. I think in order to make such a claim about the power of high-quality word embeddings you should include numbers comparing them to randomly initialized embeddings.

Questions
- Did you find that your results were sensitive to the initialization technique used for your random LSTMs / projections?
- Do you have a sense of why random non-linear features are able to perform well on these tasks? What kind of features are the skip-thought and InferSent representations learning if they do not perform much better? It’s interesting that many of the random encoder methods outperform the trained models on word content. I think you could discuss these Analysis section findings more.

Other Critiques
- In the introduction, instead of simply describing what is commonly done to obtain and evaluate sentence embeddings, it would be better to include a sentence or two about the motivation for sentence embeddings at all.
- The first sentence, “Sentence embeddings are learned non-linear recurrent combinations of pre-trained word embeddings”, doesn’t seem to be true as BOE representations are also sentence embeddings and CNNs/transformers could also work. “Non-linear” and “recurrent” are not inherent requirements for sentence embeddings, but just techniques that researchers commonly use.
- In the second paragraph of introduction instead saying “Natural language processing does not yet have a clear grasp on the relationship between word and sentence embeddings…” it might be better to say “NLP researchers” or the “NLP community” instead of “NLP” as a field doesn’t have a clear grasp.
- In the introduction: “It is unclear how much sentence-encoding architectures improve over the raw word embeddings, and what aspect of such architectures is responsible for any improvement.” It would be also good to mention that it’s unclear how much the training task / procedure also is affects improvements.
- You could describe more about applications of reservoir computing in your related work section as it’s been used in NLP before.
- I don’t think you actually ever describe the type of data that InferSent is trained on, only that it is “expensive” annotated data. It might be useful to add a sentence about natural language inference for clarity.
- In the conclusion, change “performance improvements are less than 1 and less than 2 points on average over the 10 SentEval tasks, respectively” to  “performance improvements are less than 2 percentage points on average over the 10 SentEval tasks, respectively”
- It would be nice if you bolded/underlined the best performing numbers in your results tables.

---

> ### Author Response · Authors · 2018-11-26
> **Thank you for all the feedback!**
>
> Thank you for all the feedback!
>
> We have softened the claims about the usefulness of some of the SentEval tasks for evaluation in our paper. We do think these tasks could be useful as evaluations in some situations, and strong performance should definitely be possible for very discriminative sentence embeddings. Our motivation for that take-away was also based on other observations  of these tasks (like that they are too sentiment-focused or that they are nearly solved) that have been brought to light in other works.
>
> We found initialization to matter somewhat in our experiments which is why we were very explicit about how we initialized in our submission, and we have since added some more analysis of this in the paper. In Appendix D, we compare six different initialization schemes (Heuristic (the one used in the paper currently, Uniform, Normal, Orthogonal, He (He et al. 2015), and Xavier (Glorot & Bengio, 2010). We found that BOREP is more robust to initialization than RandLSTM and prefers Orthogonal initialization. RandLSTM performs poorly with Normal initialization (and also Uniform but to a much lesser degree), and seems to perform best with He initialization.
>
> Your idea about using random word embeddings is very interesting! In fact, we added this experiment in the newest version of our paper. We included an analysis of completely random word embeddings along with the random architectures. Like in the initialization experiments, we experimented with six different methods to initialize both the word embeddings and the parameters of the architectures. The experiments are in Appendix E. We also experimented with pooling 4096 dimension embeddings (randomly sampled), which performs very well compared BOREP and RandLSTM. Overall, it really depends on the task how much the pre-trained embeddings help. However, they do seem to help more for tasks measuring semantic similarity (which makes sense since they can make use of unseen embeddings if both unseen embeddings are in the two sentences being compared). For some tasks like MRPC or SICK-E, the difference between using random word embeddings or pretrained ones is small (0.6 and 0.7 respectively), but for others like SST2 or MR it can be pretty large (10 and 8.8 points respectively). The average gain across tasks is 5.4 points.
>
> We agree that we should have added some more analysis, and we have done so in the latest version of the draft. It's difficult to say what general knowledge IS and ST models have learned and how applicable it is for the downstream tasks. This is the motivation for probing tasks (Adi et al. 2017, Conneau et al. 2018) which help measure this to a degree and show that IS and ST are able to better capture sequential information. Random networks do about as well as these pretrained encoders on tasks that can be solved just based on word content. Therefore, if the downstream tasks rely mostly on word content (or perhaps that and a type of sequential information that is not learned by IS or ST), we would expect the difference between a random encoder and IS/ST to be small.
>
> Thank you for your other critiques. We have addressed all of these in the newest version of the paper.

---

### Public Comment · (anonymous) · 2018-11-13
**Relation with LSH**

Is there any relationship with Locality Sensitive Hashing?

---

> ### Author Response · Authors · 2018-11-27
> **There is some relation.**
>
> There is some relation. LSH is collection of methods for dimension reduction and is often used for clustering. In contrast, we are increasing the dimension of embeddings in order to provide more features for downstream tasks.

---

### Author Response · Authors · 2018-11-26
**Updated version**

To all reviewers:

Thank you so much for your feedback. We have made both minor (but important) and more substantial improvements to the paper due to the thoughtful feedback you have provided. These larger improvements include experiments with different random initializations for BOREP and RandLSTM; experiments for BOREP, BOE (300 dim), BOE (4096 dim), and RandLSTM with different initializations and random word embeddings; adding standard deviations to the experimental results (and also bolded the top numbers to make interpreting the tables easier); adding more analysis regarding the probing experiments; and we included a detailed analysis of how max pooling over padding affected reported results in various papers. We hope that you find the paper improved and a more interesting read. Thank you again for your comments.

---

### Meta-Review · Area_Chair1 · 2018-12-13
**Limited but worthwhile contribution**

**Confidence:** 4
**Recommendation:** Accept (Poster)

**Metareview:**

This paper provides a new family of untrained/randomly initialized sentence encoder baselines for a standard suite of NLP evaluation tasks, and shows that it does surprisingly well—very close to widely-used methods for some of the tasks. All three reviewers acknowledge that this is a substantial contribution, and none see any major errors or fatal flaws.

One reviewer had initially argued the experiments and discussion are not as thorough as would be typical for a strong paper. In particular, the results are focused on a single set of word embeddings and a narrow class of architectures. I'm sympathetic to this concern, but since there don't seem to be any outstanding concerns about the correctness of the paper, and since the other reviewers see the contribution as quite important, I recommend acceptance. [Update: This reviewer has since revised their review to make it more positive.]

(As a nit, I'd ask the authors to ensure that the final version of the paper fits within the margins.)